



# Fast and accurate learned multiresolution dynamical downscaling for precipitation

Jiali Wang[1], Zhengchun Liu[2], Ian Foster[2], Won Chang[3], Rajkumar Kettimuthu[2], and V. Rao Kotamarthi[1]

[1]Environmental Science Division, Argonne National Laboratory, Lemont, IL, USA
[2]Data Science and Learning Division, Argonne National Laboratory, Lemont, IL, USA
[3]Division of Statistics and Data Science, University of Cincinnati, Cincinnati, OH, USA

**Correspondence:** V. Rao Kotamarthi (vrkotamarthi@anl.gov); Zhengchun Liu (zhengchun.liu@anl.gov)

**Abstract.** This study develops a neural network-based approach for emulating high-resolution modeled precipitation data with comparable statistical properties but at greatly reduced computational cost. The key idea is to use combination of low- and high-resolution simulations to train a neural network to map from the former to the latter. Specifically, we define two types of CNNs, one that stacks variables directly and one that encodes each variable before stacking, and we train each CNN type both with a
conventional loss function, such as mean square error (MSE), and with a conditional generative adversarial network (CGAN), for a total of four CNN variants. We compare the four new CNN-derived high-resolution precipitation results with precipitation generated from original high resolution simulations, a bilinear interpolater and the state-of-the-art CNN-based super-resolution (SR) technique. Results show that the SR technique produces results similar to those of the bilinear interpolator with smoother spatial and temporal distributions and smaller data variabilities and extremes than the original high resolution simulations.
While the new CNNs trained by MSE generate better results over some regions than the interpolator and SR technique do, their predictions are still not as close as the original high resolution simulations. The CNNs trained by CGAN generate more realistic and physically reasonable results, better capturing not only data variability in time and space but also extremes such as intense and long-lasting storms. The new proposed CNN-based downscaling approach can downscale precipitation from 50 km to 12 km in 14 min for 30 years once the network is trained (training takes 4 hours using 1 GPU), while the conventional
dynamical downscaling would take 1 month using 600 CPU cores to generate simulations at the resolution of 12 km over contiguous United States.

## 1 Introduction

Earth system models (ESMs) integrate the interactions of atmospheric, land, ocean, ice, and biosphere and generate principal data products used across many disciplines to characterize the likely impacts and uncertainties of climate change (Heavens
et al., 2013; Stouffer et al., 2017). The computationally demanding nature of ESMs, however, limits their spatial resolution mostly to between 100 and 300 km. Such resolutions are not sufficient to resolve critical physical processes such as clouds, which play a key role in determining the Earth's climate by transporting heat and moisture, reflecting and absorbing radiation, and producing rain. Moreover, ESMs cannot assess stakeholder-relevant local impacts of significant changes in the attributes of these processes (at scales of 1–10 km; Gutowski Jr et al., 2020). Higher-resolution simulations covering the entire globe





are emerging (e.g., Miyamoto et al., 2013; Bretherton and Khairoutdinov, 2015; Yashiro et al., 2016), including the U.S. Department of Energy's 3 km Simple Cloud-Resolving E3SM Atmosphere Model (E3SM Project, 2018). They are expected to evolve relatively slowly, however, given the challenges of model tuning and validation as well as data storage at unfamiliar scales (Gutowski Jr et al., 2020).

Downscaling techniques are therefore used to mitigate the low spatial resolution of ESMs. Figure 1 illustrates several ap-
proaches. Statistical downscaling is computationally efficient and thus can be used to generate multimodel ensembles that are generally considered to be required for capturing structural and scenario uncertainties in climate modeling (Hawkins and Sutton, 2009; Deser et al., 2012; Mearns et al., 2012; Mezghani et al., 2019). However, statistical downscaling works only if the statistical relationship that is calibrated with the present climate is valid for future climate conditions (Fowler et al., 2007). This "stationarity assumption" cannot always be met in practice (Wang et al., 2018). In addition, typical statistical downscaling is
limited by the availability of observations, which may lack both spatial and temporal coverage. Furthermore, observations may contain errors, posing challenges for developing a robust model to project future climate.

Dynamical downscaling, in contrast, uses ESM outputs as boundary conditions for regional climate model (RCM) simulations to produce high-resolution outputs. The RCM typically resolves atmospheric features at a spatial resolution of 10–50 km (depending on factors such as the size of the studied domain) with parameterized physical atmospheric processes that in many
cases are similar to those used in the ESMs. This approach has the value of being based on physical processes in the atmosphere (e.g., convective scheme, land surface process, short/long wave radiation) and provides a description of a complete set of variables over a volume of the atmosphere. Running an RCM is computationally expensive, however, and typically cannot be applied to large ESM ensembles, especially when simulating at the high spatial resolution required to explicitly resolve the convection that cause precipitation storms. For example, a 30-year simulation over a region covering the central and eastern
United States with the Weather Research and Forecasting (WRF V4.1.3) model takes 4380 core-hours at 50 km resolution but 374,490 core-hours at 12 km resolution and 5.4 million core-hours at 4 km resolution (i.e., 1,224 times more computing resource than at 50 km) on the Intel Broadwell partition of the Bebop cluster at Argonne National Laboratory.

Another recently proposed approach to downscaling uses deep neural networks (DNNs), specifically DNN-based super-resolution (SR) techniques. A DNN consists of several interconnected layers of nonlinear nodes with weights determined by a
training process in which the desired output and actual output are repeatedly compared while weights are adjusted (LeCun et al., 2015). DNNs can approximate arbitrary nonlinear functions and are easily adapted to novel problems. They can handle large datasets during training and, once trained, can provide fast predictions. In digital image processing, DNN-based SR (Dong et al., 2014; Yang et al., 2014) describes various algorithms that take one or more low-resolution images and generate an estimate of a high-resolution image of the same target (Tian and Ma, 2011), a concept closely related to downscaling in
climate modeling. They employ a form of DNN called a convolutional neural network (CNN; LeCun et al., 1998) in which node connections are configured to focus on correlations within neighboring patches. Another DNN variant, the generative adversarial network (GAN; Goodfellow et al., 2014), has been used to improve the accuracy of super-resolution CNNs (Ledig et al., 2017).





SR methods have recently been applied to the challenging problems of downscaling precipitation (Vandal et al., 2017; Geiss
and Hardin, 2019) and wind and solar radiation (Stengel et al., 2020), quantities that can vary sharply over spatial scales of
10 km or less depending on location. Downscaling with an SR model proceeds as follows (Vandal et al., 2017; Stengel et al.,
2020): (1) take high-resolution data (either climate model output or gridded observations), upscale the data to a low resolution;
(2) build an SR model using the original high-resolution and the upscaled low-resolution data; and (3) apply the trained SR
model to new low-resolution data, such as from an ESM, to generate new high-resolution data. This general approach has
achieved promising results but also has problems. The trained SR model often performs well when applied to a low-resolution
data (upscaled from high-resolution data) and compared with the same high-resolution data, capturing both spatial patterns and
sharp gradients (Geiss and Hardin, 2019), especially when using a GAN (Stengel et al., 2020). This result is not surprising,
given that the high- and low-resolution data used to develop the SR are from the same source. When applied to new low-
resolution data such as from an ESM, however, the SR may generate plausible-looking fine features but preserves all biases
that exist in the original ESM data, especially biases in spatial distributions and in time series such as diurnal cycles of
precipitation, both of which are important for understanding the impacts of heavy precipitation.

Here we describe a new **learned multiresolution dynamical downscaling** approach that seeks to combine the strengths of
the dynamical and DNN-based downscaling approaches. Two datasets rather than one are used to develop the new CNN-based
downscaling approach. As shown in Figure 1, these two datasets are both generated by dynamical downscaling with an RCM
driven by the same ESM as boundary conditions but at different spatial resolutions. We then use the CNN to approximate
the relationship between these two datasets, rather than between the original and upscaled versions of the same data as the
SR downscaling does. The output of the CNNs is expected to generate fine-scale features as in the original high-resolution
data, because the algorithm is built based on both high and low resolutions. Our goal in developing this approach is to enable
generation of fine-resolution data (e.g, 12 km in this study) based on relatively coarse-resolution RCM output (e.g., 50 km in
this study) with low computational cost. The combination of high computational efficiency and high-resolution output would
allow building more robust datasets for meeting stakeholder needs in infrastructure planning (e.g., energy resource, power
system operation) and policy making, where higher spatial resolution (1–10 km) is usually desired. In contrast to statistical
downscaling, this approach does not rely on any observations; thus, it can downscale any variables of interest from RCM output
for different disciplines. Moreover, because the approach is built on dynamically downscaled simulations, these datasets are
not bound by stationarity assumptions.

## 2   Data and Method

This study focuses on precipitation, which is highly variable in time and space and is often the most difficult to compute in
ESMs (Legates, 2014). Downscaling of ESMs with RCMs has generally reduced the bias in precipitation projections, often due
to an increase in the model spatial resolution that allows for resolving critical terrain features such as changes in topography and
coastlines (Wang et al., 2015; Zobel et al., 2018; Chang et al., 2020). In addition, precipitation data at high spatial resolution is
needed for a variety of climate impact assessments, ranging from flooding risk to agriculture (Maraun et al., 2010; Gutowski Jr



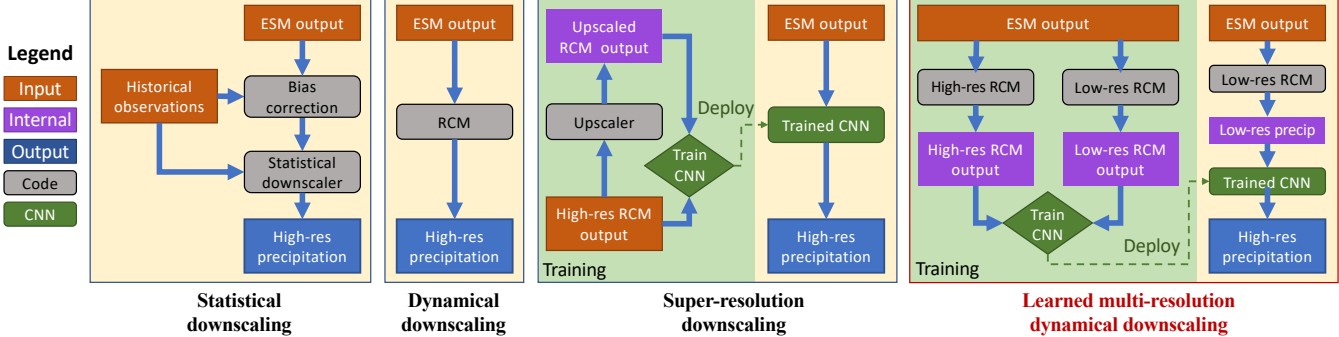

**Figure 1.** Four downscaling approaches discussed in the text. Orange and blue rectangles are input and output, respectively, of each downscaling approach, and grey rounded rectangles are computational steps. In our learned multiresolution dynamical downscaling approach (right), we use the outputs from low- and high-resolution dynamical downscaling runs driven by the same ESM as boundary conditions to generate {Low, High} training data pairs for a CNN that, once trained, will map from low-resolution dynamically downscaled outputs to a high-resolution.

et al., 2020). Precipitation data produced by RCMs at each model timestep can be viewed as a two-dimensional matrix or image. However, these precipitation images are different from typical photographic images. For example, the precipitation generated by dynamical downscaling at low and high spatial resolutions can be different even if the RCMs used to generate

them differ only in spatial resolution. This situation appears often in the precipitation data produced by RCMs running at different spatial resolutions using the same initial and boundary conditions, and it poses a great challenge for developing DNNs for downscaling. In the following subsections we describe in detail the dataset we used for our study, and we discuss our deep learning methods.

## 2.1 Dataset

The data used in this study are one-year outputs from two RCM simulations using the Weather Research and Forecasting model version 3.3.1, one at 50 km resolution and one at 12 km resolution, both driven by National Centers for Environmental Prediction-U.S. Department of Energy Reanalysis II (NCEP-R2) for the year 2005. These two simulations were conducted separately, not in nested domains, with output every 3 hours for a total of 2920 timesteps in each dataset. Our study domain covers the contiguous United States (CONUS), with 512×256 grid cells for the 12 km simulation and 128×64 grid cells for

the 50 km simulation.

The two WRF simulations share the same configuration and physics parameterizations; they differ only in their spatial resolutions. This difference has two direct effects on the precipitation pattern. One is that the higher-resolution results can better resolve physical processes when compared with lower resolution. For example, we can expect the high-resolution simulation to have improved performance for processes that are scale dependent, such as convections and planetary boundary layer physics

(Prein and Giangrande, 2020). The other effect is that the higher-resolution model resolves terrain and hence terrain-influenced



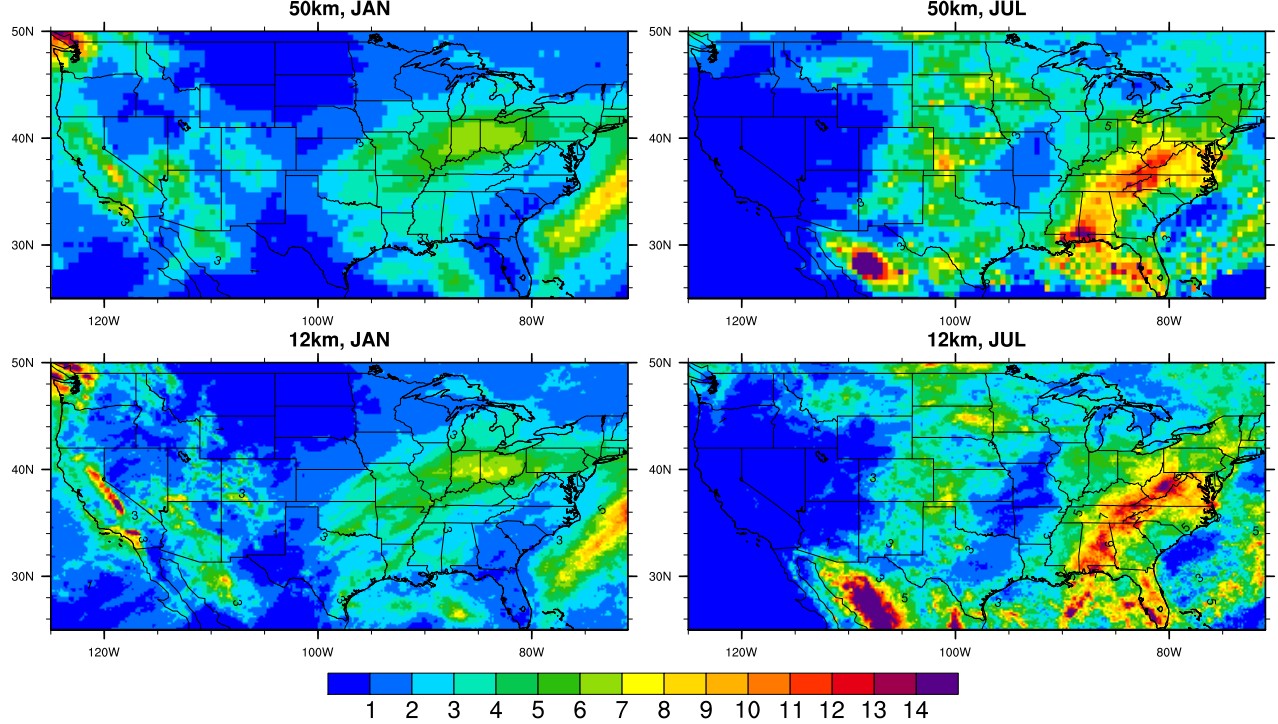

**Figure 2.** Averaged daily precipitation (mm/day) in January and July of 2005 using 50 km (top) and 12 km (bottom) WRF output. The 50 km precipitation data not only miss fine-scale features, especially over complex terrain such as Sierra Nevada and Appalachians shown by 12 km, but also generate precipitation in different locations, such as July precipitation over Texas.

rainfall, land-sea interface, and coastal rainfall better than the coarse-resolution model does (Komurcu et al., 2018). The difference in spatial resolution also has indirect effects on the precipitation pattern. For example, because these two simulations are not nested, they cover slightly different domains, even though they maintain the same region (CONUS) in the interior. This minor difference in domain position can change the large-scale environment and translate into diverse conditions for the

development of the mesoscale processes that produce precipitation (Miguez-Macho et al., 2004). In addition, the difference in spatial resolution leads to the two simulations using different computing timesteps (120 sec versus 40 sec), which can cause precipitation differences due to operator splitting between dynamics and physics in the WRF (Skamarock et al., 2005; Skamarock and Klemp, 2008; Barrett et al., 2019). These factors lead to precipitation differences between the 50 km and 12 km WRF output, as seen in Figure 2, which shows differences between these two datasets in daily January and July means. The

50 km and 12 km data not only have different fine-scale features, such as in the Sierra Nevada and Appalachians, but also have different geolocations of the precipitation, such as that over Texas in July, where the 50 km simulation produces precipitation for 3–5 mm/day but the 12 km simulation produces only 1–2 mm/day. The difference in precipitation images between low and high resolution is the biggest challenge for our proposed downscaling approach.





**Table 1.** Inputs and outputs for the new CNNs developed in this study. The range column shows the top and bottom 0.1% of the variable over the study domain for the time series of 3-hourly data in year 2005. All data are produced by the WRF model.

| Input (units) | Range |
| --- | --- |
| 50 km, 3-hourly precipitation (mm/3 hr) | [0.05, 13.62] |
| 50 km, 3-hourly SLP (hPa) | [990.97, 1039.34] |
| 50 km, 3-hourly IWV (cm) | [1.56, 116.46] |
| 50 km, 3-hourly T2 (K) | [241.75, 310.35] |
| 12 km, topographic height (m) | [0, 3204.51] |
| Output | |
| 12 km, 3-hourly precipitation (mm/3 hr) | [0.05, 15.66] |

Given these datasets, we need to decide which variables to provide as inputs to our DNN-based downscaling system. Many
factors influence the magnitude and variability of precipitation, the focus of this study. Informed by the physics of precipitation,
we include, in addition to the low-resolution precipitation and high-resolution topography data used by Vandal et al. (2017)
for their SR model, the vertically integrated water vapor (IWV) or precipitable water, sea level pressure (SLP), and 2-meter air
temperature (T2) as inputs (Table 1) since we find these variables show high pattern correlations with precipitation along the
time dimension. Each variable possesses rich spatial dependencies, much like images, although climate data are more complex
than images because of their sparsity, dynamics, and chaotic nature. For each grid cell, we use a fixed threshold (0.05 mm/3
hr) for the minimum precipitation amount, so as to avoid zeros and drizzles being passed to the neural network. Similarly, we
define a 99.5th percentile for the maximum precipitation over each grid cell, so as to avoid extremely large precipitation values
being passed to the neural network. Our proposed DNN-based downscaling technique is different from the traditional statistical
downscaling methods, particularly regression-based models, which vectorize spatial data and remove the spatial structure.

**2.2 Stacked Variables**

We design different model architectures and loss functions to make best use of the input variables when training the CNNs
to capture the relationship between the precipitation generated by the low- and high-resolution simulations. In this case, we
directly stack all selected variables (precipitation, terrain height, T2, IWV, SLP) to form a three-dimensional tensor as input
to the CNN model; see Figure 3. We call the resulting method Direct-Simple hereafter. This approach of treating the climate
variables as images and directly stacking them as different channels has been used in other downscaling studies (e.g., Vandal
et al., 2017). Unlike those previous studies, however, we use an inception module as a building block because it can provide
different receptive fields at each layer (Szegedy et al., 2015). We use kernels of size 1×1, 3×3, and 5×5 (Figure 4) to build
the inception module in order to mitigate the challenge of learning the relationship between the low- and high-resolution
simulations when precipitation occurs in different locations in the two datasets.



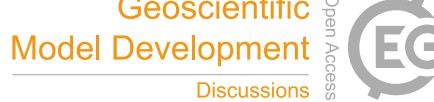

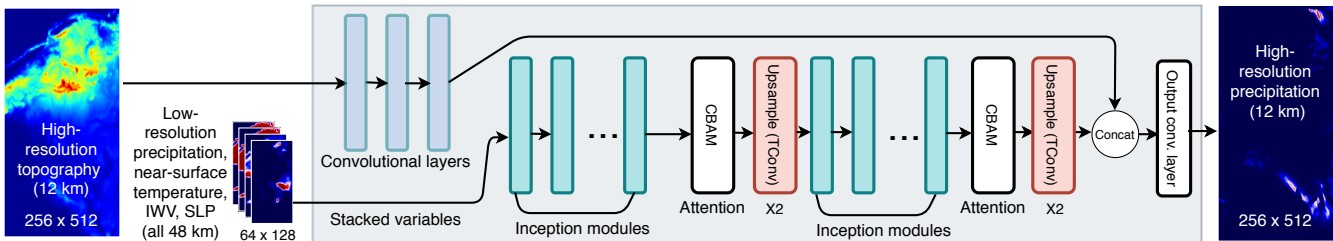

**Figure 3.** Model architecture for Direct-Simple and the generator of Direct-CGAN. CBAM=Convolutional Block Attention Module (Woo et al., 2018). TConv=Transposed convolution. The inception module is shown in Figure 4

.

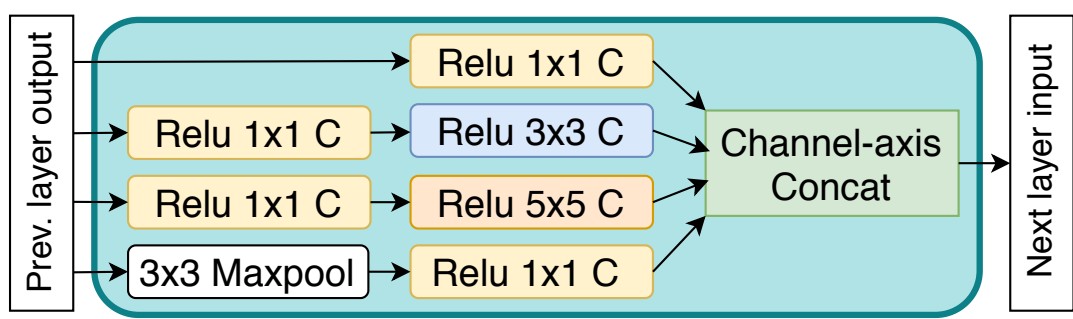

**Figure 4.** Details of the inception module used in Figure 3.

From a physical perspective, the inception module makes sense because the precipitation at a location or area is influenced by the conditional variables not only at that particular location but also at adjacent locations depending on the types of weather system. For example, precipitation associated with tropical cyclones (with low SLP centers) over the southeastern United States is usually produced at the eastern or northeastern side of the cyclone center, where the moisture is brought from the Atlantic Ocean or Gulf of Mexico to the northern inland. In addition, stacking the variables considers the coupling effect of

all the variables that simultaneously influence the occurrence of precipitation but whose relative importance can be different. Therefore, we apply channel attention (Woo et al., 2018) so that the CNN can learn to focus on the important physical factors that influence the precipitation. On the other hand, the precipitation is extremely sparse in space, with many zeros, posing challenges for the training process. To account for the sparsity of data, we apply a spatial attention (Woo et al., 2018) mechanism to allow the model to learn to emphasize or suppress and refine intermediate features effectively so as to focus on important

areas with relatively large precipitation values. This makes physical sense because capturing large precipitation events is critical for climate impact applications, more so than are drizzle or no-precipitation days.



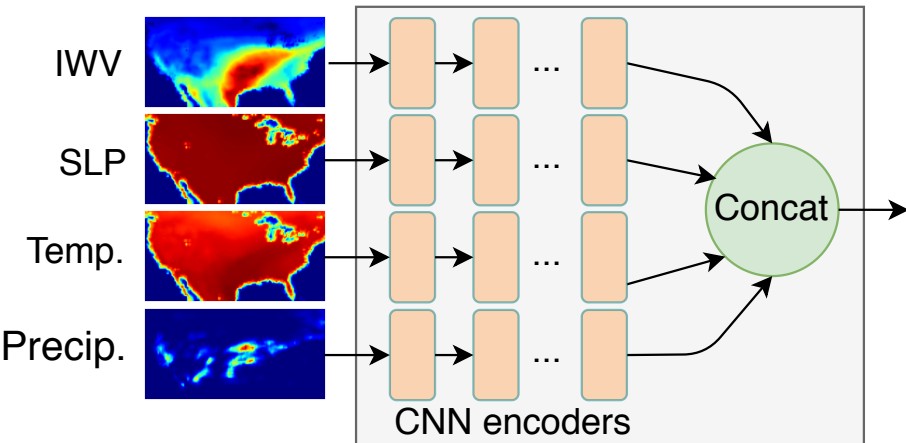

**Figure 5.** Encoder module used in the Encoded-Simple and the generator of Encoded-CGAN models to prepare the low-resolution input data prior to passing the data to the network of Figure 3.

## 2.3 Encoded Variables

Stacking all variables as different channels in a CNN is simple and straightforward. However, combining these variables that are significantly different in scales, distribution shapes, sparsity, and units as shown by Table 1 can make the training

process challenging. Thus, we develop an encoded variable CNN in which dedicated convolution layers are provided for each variable to extract features before stacking. This ensures that when we stack, the feature maps extracted from each variable have similar characteristics. We call the resulting method Encoded-Simple hereafter and refer to Direct-Simple and Encoded-Simple collectively as the *Simple models*.

Specifically, as shown in Figure 5, we design convolution layers for each of the four variables and stack (i.e., concatenate in the channel axis) their feature maps. We process the topography data similarly but stack the feature maps after the second

upsampling operation, that is, when the feature maps size becomes $512 \times 256$. From a physics perspective, Encoded-Simple is similar to Direct-Simple except that there are dedicated convolution layers to learn to extract features from each variable during the training process. From the deep learning perspective, this process is friendlier to training, and thus we expect better results from Encoded-Simple than from Direct-Simple. In this study we consider spatial attention for the feature map of each variable

before stacking them (i.e., concatenate in the channel axis) and then channel attention (the relative importance of different input variables), because the spatial feature of precipitation is sparse and less uniform and is a critical factor for judging the model performance.

## 2.4 Super-Resolution Model

To compare the performance of our proposed learned multiresolution dynamical downscaling approach with that of the state-

of-the-art SR technique, we develop an SR model based on the original 12 km WRF modeled precipitation and an upscaled





12 km-to-50 km dataset. The SR model development does not need other environmental variables as used in our new CNN approaches. It does not involve the 50 km precipitation data either, thus, the differences between the 12 km and 50 km WRF outputs (as discussed earlier and shown in Figure 2) is not a challenge in the way that it was for the learned multiresolution dynamical downscaling approach. We then (1) apply the trained SR model to the upscaled 50 km dataset and compare the SR-

resulted 12 km data with the original 12 km data. This step is to assess the effectiveness of the SR model. For example, when comparing SR-resulted 12 km data and original 12 km data, we find a spatial correlation greater than 0.98 for all the quantiles and almost the same distribution shapes between the two datasets, indicating that the SR model we develop is effective and robust; (2) apply the SR model to our 50 km WRF output for the testing period and compare the resulting SR-generated 12 km data with the original 12 km data. If the SR model can downscale a 50 km dataset to one with similar properties to 12 km WRF

output, with much less computational cost than running the 12 km model in conventional dynamical downscaling, then the SR approach is useful for generating high-resolution and high-fidelity precipitation based on low-resolution precipitation

## 2.5 Loss Functions

A DNN's loss function guides the optimization process used to update weights during training. Thus the choice of loss function is crucial to DNN effectiveness. For the Direct-Simple and Encoded-Simple models introduced above, we first consider two

loss functions commonly used in computational vision: the L1 norm (mean absolute error) and L2 norm (mean square error: MSE). Since precipitation data are sparse, those losses may not be able to generate results that are driven primarily by large gradients, such as localized heavy precipitation.

The generative adversarial network (GAN) is a class of machine learning framework in which two neural networks, generator and discriminator, contend with each other to produce a prediction. In our context, the generator network performs CNN by

mapping input patches of coarse data to the space of the associated higher-resolution patches. The discriminator attempts to classify proposed patches as real (i.e., coming from the training set) or fake (i.e., coming from the generator network). The two networks are trained against each other iteratively, and over time the generator produces more realistic fields, while the discriminator becomes better at distinguishing between real and fake data. Therefore, GANs provide a method for inserting physically realistic, small-scale details that could not have been inferred directly from the coarse input images.

GANs, as originally formulated, use a vector of random numbers (latent variables) as the only input to the generator. Instead, we use actual precipitation amounts and the conditional variables as inputs, forming a conditional GAN (CGAN) framework (Mirza and Osindero, 2014) for training the generator. The two neural networks (generator and discriminator) are trained simultaneously, with the generator using a weighted average (their weights are hyperparameters) of the $\ell_1$-norm (results with the $\ell_2$-norm are less good) and adversarial loss as its loss function, defined as

$$\ell(\theta_G) = -\frac{w_a}{m} \sum_{i=1}^{m} D\left(G\left(v_1, v_2, \ldots\right)\right) + \frac{w_c}{m} \left\|Y_i - \overline{Y}_i\right\|_1, \qquad (1)$$

where $w_a$ and $w_c$ are weights for the adversarial loss and $\ell_1$-norm, respectively; $m$ is the minibatch size; $D$ is the discriminator; $G$ is the generator; $\theta_G$ denotes the trainable parameters (i.e., weights) of the generator; $v_1, v_2, \ldots$ are input variables; $Y$ is the precipitation at time grid cell $i$; and $\overline{Y}_i$ is the regional average precipitation. Based on a hyperparameter study, we selected





values of 1 and 5 for $w_a$ and $w_c$, respectively. The discriminator loss is

$$\ell(\theta_D) = \frac{1}{m} \sum_{i=1}^{m} [D(G(v_1, v_2, \ldots) - D(P_h)], \tag{2}$$

where $P_h$ is the corresponding high-resolution precipitation. Therefore, in addition to the L1-norm loss used to retain low-frequency content in the images, our target generator is trained to generate high-resolution precipitation patterns that are indistinguishable from the real high-resolution precipitation patterns generated by the discriminator. Once trained, only the generator is used for downscaling low-resolution precipitation data. We incorporate CGAN into both Direct-Simple and Encoded-Simple to obtain two new models that we refer to respectively as Direct-CGAN and Encoded-CGAN hereafter. We also use CGAN (with actual precipitation amounts) for training the SR model, producing a model described earlier and that we refer to as SR-CGAN.

## 2.6 Implementation and Model Training

We implement our model with the PyTorch machine learning framework. We use 3-hourly data from January to September of the year 2005 for model training and validation (e.g., hyperparameter tuning to control overfitting) and the remaining three months (October, November, December) for testing the model performance. The Adam optimizer and one NVIDIA V100 GPU are used for model training. The training is computationally intensive, taking for example about 4 hours for Encoded-CGAN on 1 NVIDIA V100 GPU for 8,000 iterations and a minibatch size of 32. Training time can be reduced by parallelism, for example to less than an hour on 8 GPUs. Once the model is trained, it takes less than a minute on one GPU to downscale three months of coarse-resolution precipitation data.

## 2.7 Evaluation Metrics

We compare the 12 km precipitation field generated by the CNN models against two different sets of WRF precipitation outputs: one from WRF run at a grid spacing of 12 km (referred to as *Ground Truth*), and a second generated by interpolating output from a 50 km run to 12 km (referred to as *Interpolator*). We use the 12 km WRF modeled precipitation as Ground Truth because the CNNs are designed to achieve the performance of these data by approximating the relationship between coarse- and fine-resolution precipitation data. We examine the statistical distribution of precipitation using MSE and the probability density function (PDF) by aggregating all the grid cells over CONUS and smaller regions. MSE is computed for each timestep for the testing period:

$$\text{MSE} = \frac{1}{N} \sum_{i=1}^{N} (Y_i - \overline{Y})^2, \tag{3}$$

where $N$ is the total number of grid cells over a domain of interest, $Y_i$ is the precipitation at grid cell $i$, and $\overline{Y}$ is the average precipitation across all the grid cells. We calculate the MSE across CONUS and also in each of seven subregions defined by the National Climate Assessment (NCA; Melillo et al., 2014), as shown in Figure 6 (lower right).





To measure the similarity between the PDFs of Ground Truth, Interpolator, and the CNN models, we employ the Jensen–Shannon (J-S) distance (Osterreicher and Vajda, 2003; Endres and Schindelin, 2003), which measures the distance between
two probability distributions. The J-S distance is computed by:

$$JSD(P,Q) = \sqrt{\frac{D(P||M) + D(Q||M)))}{2}}, \qquad (4)$$

where $P$ and $Q$ are the two probability distributions to be evaluated (i.e., distribution of RCM simulated 12 km and CNN-downscaled 12 km precipitation, respectively), $M$ is the mean of $P$ and $Q$, and $D$ is the Kullback–Leibler divergence (Kullback and Leibler, 1951) calculated with

$$D(P||M) = \sum_{x \in X} P(x) log \left( \frac{P(x)}{M(x)} \right), \qquad (5)$$

where $x$ is the bin we apply for the PDFs; here $x = 0, 1, 2, ..., 30$. We calculated the J-S distance between PDFs of the five CNN predictions and Ground Truth, with small distance indicating that the CNN predictions have similar distributions to Ground Truth. The J-S distance takes into account not only the median or the mean but also the entire distribution including the scale and the tails, which is important because variability changes are equally important especially for threshold-defined extremes,
whose frequency is more sensitive to changes (Vitart et al., 2012).

We also investigate the geospatial pattern of mean, standard deviation, and extreme values of precipitation over time. To measure whether the CNN models capture the spatial variability in these values of Ground Truth, we calculate the pattern correlations between each pair of the data, namely, Ground Truth versus Interpolator and Ground Truth versus each of the CNN models. Higher correlation indicates that the spatial variability in Ground Truth due to local effects (e.g., terrain) and
synoptic-scale circulations are captured well by Interpolator and the CNN models.

While these metrics examine precipitation either at the level of individual model grid cells or at a regional scale by aggregating all the information together into one metric, they cannot determine model performance in rainstorm characteristics such as frequency, duration, intensity, and size of individual events. This information is all confounded in local or spatially aggregated time series. To overcome this limitation, we use a feature-tracking algorithm developed for rainstorm objects in
particular (fully described in Chang et al., 2016), and we identify events using information from the precipitation field only. The algorithm applies almost-connected component labeling in a four-step process to reduce the influence of the chaining effect and allow grouping of physically reasonable events. The algorithm accounts for splits in individual events during their evolution and does not require that all precipitation be contiguous. In principle, such methods can decompose precipitation bias and distinguish between biases in the duration, intensity, size, and number of events. The duration of each event, in units
of timesteps, is

$$D = T_e - T_b + 1, \qquad (6)$$

where $T_b$ and $T_e$ are the beginning and ending timestep, respectively. The lifetime mean size $S_{life}$ for an individual precipitation event is calculated as the sum of all the area (in km$^2$; derived by number of grid cells$\times$144) associated with an event over its





lifetime divided by $D$. The lifetime mean precipitation intensity is calculated by

$$I_{life} = \frac{V_{tot}}{S_{life}},$$
(7)

where $V_{tot}$ is the total precipitation volume (in m$^3$; derived by amount×144000) over the event lifetime.

## 3 Results

We now evaluate the efficacy of the five CNN methods by comparing their predictions with the original WRF output at a grid spacing of 12 km (Ground Truth), the output of a 12 km bilinear interpolation from the 50 km data (Interpolator), and the state-
of-the-art SR-CGAN model output. We expect some of the CNN models developed by this study to generate more accurate results than Interpolator and SR-CGAN when using the same coarse-resolution RCM-modeled precipitation as input, because our CNN models are developed to approximate the relationship between the coarse- and the fine-resolution precipitation data.

### 3.1 MSE

Table 2 summarizes the MSE comparing the CNN-predicted precipitation with Ground Truth and Interpolator along the testing
period. The SR-CGAN model in general shows similar MSE to that of Interpolator, while the Simple models show smaller MSEs than that of Interpolator. There are many timesteps and regions for which Direct-CGAN and Encoded-CGAN show larger MSEs than Interpolator and Simple models. The reason is that the Simple models are trained specifically to optimize this content-based loss, resulting in fields that are safer——that is, overly smoothed predictions of high-resolution precipitation. The CGAN models, in contrast, change the landscape of the loss function by adding the adversarial term to the L1-norm
(Equation 1), which is more physically consistent with the training data, and by inserting significantly more small-scale features that better represent the nature of the true precipitation fields. However, these features also cause the high-resolution fields to deviate from Ground Truth in an MSE sense since they cannot be inferred from the low-resolution input.

### 3.2 Probability Density Function

To better validate that Direct-CGAN and Encoded-CGAN have learned the appropriate distribution for the precipitation data,
we assess the PDFs of precipitation for the five CNN models across all timesteps over the entire CONUS area and the seven subregions. For a certain region, we take into account all the grid cells and the timesteps (without any averages in space or time) for the density function. As shown in Figure 6, while all results show similar probability densities for small and moderate precipitation, Ground Truth usually has longer tails than Interpolator and SR-CGAN have. SR-CGAN shows almost identical distribution to Interpolator, with smaller densities for the large precipitation than Ground Truth, and larger J-S distance from
Ground Truth compared with the four new CNN models developed here. According to the J-S distance (Table 3), Direct-Simple and Encoded-Simple show closer distributions to Ground Truth than Interpolator and SR-CGAN do over some subregions such as Southwest, Northeast, Southern Great Plains, and Northwest, but they are still similar to or even worse than Interpolator over other regions, underestimating the heavier precipitation over the Northern Great Plains, Midwest, and Southeast. Direct-CGAN





**Table 2.** MSEs, calculated across all grid cells over the entire CONUS and seven subregions (Equation 3), at the 50th and 99th percentiles picked from all timesteps. The MSEs of the CGAN models are not necessarily smaller than those of the Simple models, especially for heavier precipitation, which has larger MSEs.

|  | CONUS | Southwest | Northeast | Midwest | S Great Plains | Northwest | N Great Plains | Southeast |
|---|---|---|---|---|---|---|---|---|
| Interpolator | 0.24, 1.07 | 0.18, 1.87 | 0.13, 3.56 | 0.10, 3.77 | 0.07, 3.91 | 0.07, 3.68 | 0.07, 3.64 | 0.07, 3.77 |
| SR-CGAN | 0.25, 1.01 | 0.19, 1.80 | 0.14, 4.24 | 0.10, 4.33 | 0.07, 4.34 | 0.08, 4.19 | 0.07, 4.11 | 0.07, 4.37 |
| Direct-Simple | 0.20, 0.96 | 0.16, 1.24 | 0.12, 3.04 | 0.10, 3.22 | 0.07, 3.39 | 0.08, 3.07 | 0.07, 3.03 | 0.07, 3.31 |
| Encoded-Simple | 0.21, 1.03 | 0.16, 1.39 | 0.12, 3.55 | 0.09, 3.75 | 0.06, 4.12 | 0.07, 3.75 | 0.07, 3.57 | 0.06, 4.06 |
| Direct-CGAN | 0.21, 1.05 | 0.16, 1.58 | 0.11, 3.78 | 0.09, 4.06 | 0.06, 4.21 | 0.07, 3.96 | 0.06, 3.86 | 0.06, 4.09 |
| Encoded-CGAN | 0.22, 1.03 | 0.17, 1.43 | 0.12, 3.57 | 0.09, 3.73 | 0.06, 3.96 | 0.07, 3.79 | 0.07, 3.77 | 0.06, 4.30 |

and Encoded-CGAN produce better precipitation distributions over these three subregions and show smaller J-S distance from Ground Truth than Interpolator, SR-CGAN, and the Simple models. This finding indicates that although Direct-Simple and Encoded-Simple obtain lower grid cell-wise error than Direct-CGAN and Encoded-CGAN do, they cannot capture the small-scale features (i.e., local extremes in time and space) that better represent the nature of the precipitation fields.

### 3.3 Geospatial Analysis of Other Measures

To investigate whether the five CNNs can capture the geospatial pattern of the mean, standard deviation, and top 5% precipitation seen in the Ground Truth, we conduct geospatial evaluations of these measures because they assess the performance of CNNs in a more accurate manner. For example, if the location of a heavy precipitation event is misrepresented in the CNN predictions, bias can be seen in geospatial maps but not in the PDFs for a specific region (Figure 6). We expect the Simple models to be smoother in space for all the quantities because they underestimate the large precipitation value but have smaller MSEs than Interpolator has. In contrast, we expect Direct-CGAN and Encoded-CGAN to show sharper gradients in space for all the quantities, especially for standard deviation and the top 5% precipitation, because the CGAN model better captures the long tails of the PDFs, which is often driven by strong data variabilities.

We start by comparing the mean state of precipitation for each dataset over all the grid cells. As shown in Figure 7, Interpolator shows a smoother precipitation pattern than Ground Truth does, with fewer large values over Northwest, where the largest precipitation occurs in the cold season. SR-CGAN shows almost the same spatial pattern as Interpolator does. The fine-scale features added by SR-CGAN do not seem to help improve the underestimation of precipitation over Northwest. The pattern correlations between Ground Truth and Interpolator and between Ground Truth and SR-CGAN are also similar (0.866 and 0.871). The Simple models generate smaller-scale features, which are especially seen over northwestern regions, and the precipitation values are also larger along the western coast than Interpolator and SR-CGAN are. The pattern correlation between Ground Truth and Direct-Simple and that between Ground Truth and Encoded-Simple are improvements over Interpolator



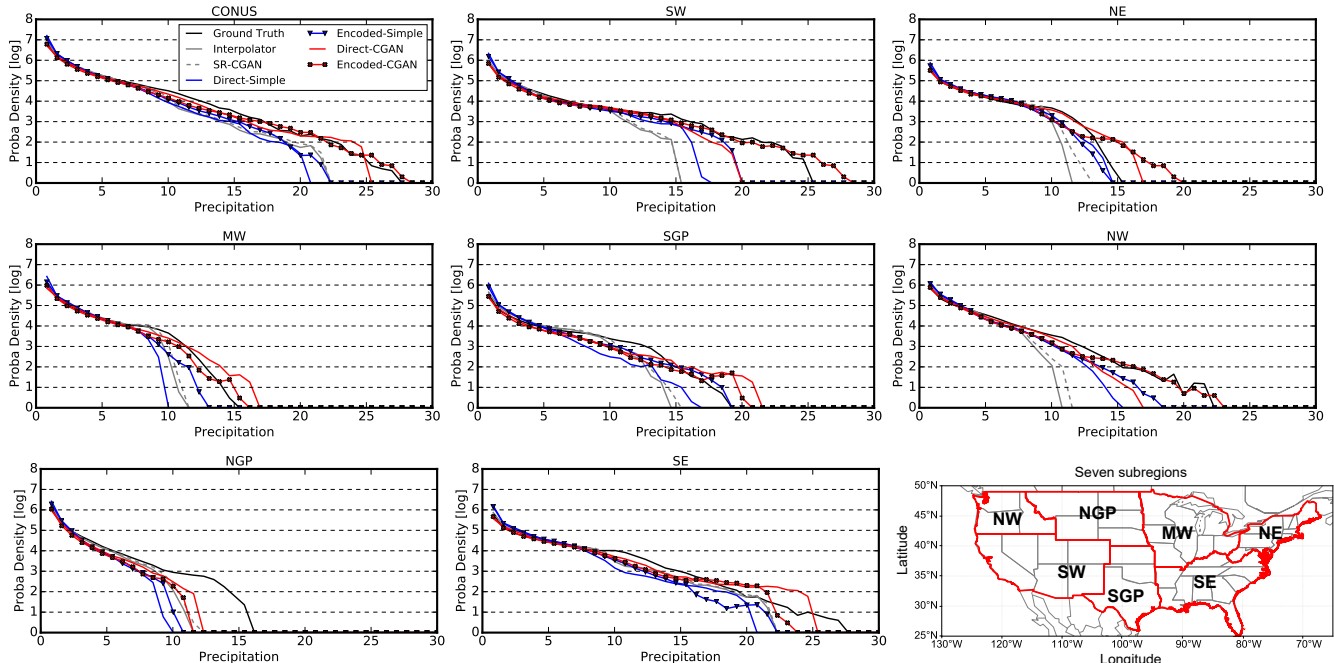

**Figure 6.** PDFs from Ground Truth, Interpolator, and CNN-predicted precipitation calculated based on grid cells and timesteps over CONUS and seven subregions. The subregions are based on those used in the national climate assessment: Northeast (NE), Southeast (SE), Midwest (MW), Southwest (SW), Northwest (NW), Northern Great Plains (NGP), and Southern Great Plains (SGP).

from 0.866 to 0.937 and 0.945, respectively. However, there is still an underestimation of high values (>1.7 mm) over Northwest. The precipitation patterns over the central and eastern United States generated by the Simple models are similar to those of Interpolator and SR-CGAN, with overestimation of light precipitation (0.1–0.5 mm) over Northern Great Plains and the southeastern states such as Louisiana and Mississippi. The Direct-CGAN shows greater precipitation than the Simple models do over Northwest; and both Direct-CGAN and Encoded-CGAN show smaller precipitation (similar to Ground Truth) than

Interpolator, SR-CGAN, and Simple models do over Great Plains and the southeastern states. The pattern correlation between Ground Truth and Direct-CGAN and that between Ground Truth and Encoded-CGAN are improved over Interpolator from 0.866 to 0.943 and 0.951, respectively. The improvements of the pattern correlation by the CGAN models indicate that they can better capture the spatial variability of the mean precipitation shown in Ground Truth than Interpolator and other CNN models do.

Next, we investigate the performance of the five CNN models for capturing higher-order statistics of precipitation, such as standard deviation and top 5% precipitation, since climate modeling is as much concerned with variability as with mean values. As shown in Figure 8, in Ground Truth the precipitation during October to December over the northwestern coast shows a standard deviation up to 2 mm/3 hr, the largest across the entire CONUS area. Standard deviation over the eastern United States is also large (1–1.7 mm), while that over Southwest is the smallest because this time period is usually very



**Table 3.** J-S distance (Equation 4) measuring the similarity of the PDFs between Ground Truth and six predictive models (Interpolator and CNN-based models) over CONUS and seven subregions.

|  | CONUS | Southwest | Northeast | Midwest | S Great Plains | Northwest | N Great Plains | Southeast |
|---|---|---|---|---|---|---|---|---|
| Interpolator | 0.164 | 0.325 | 0.241 | 0.229 | 0.210 | 0.372 | 0.275 | 0.151 |
| SR-CGAN | 0.162 | 0.325 | 0.174 | 0.219 | 0.187 | 0.348 | 0.255 | 0.150 |
| Direct-Simple | 0.200 | 0.275 | 0.069 | 0.293 | 0.149 | 0.247 | 0.336 | 0.188 |
| Encoded-Simple | 0.166 | 0.208 | 0.083 | 0.149 | 0.038 | 0.168 | 0.305 | 0.160 |
| Direct-CGAN | 0.081 | 0.209 | 0.138 | 0.130 | 0.152 | 0.202 | 0.239 | 0.103 |
| Encoded-CGAN | 0.039 | 0.107 | 0.187 | 0.069 | 0.115 | 0.060 | 0.271 | 0.125 |

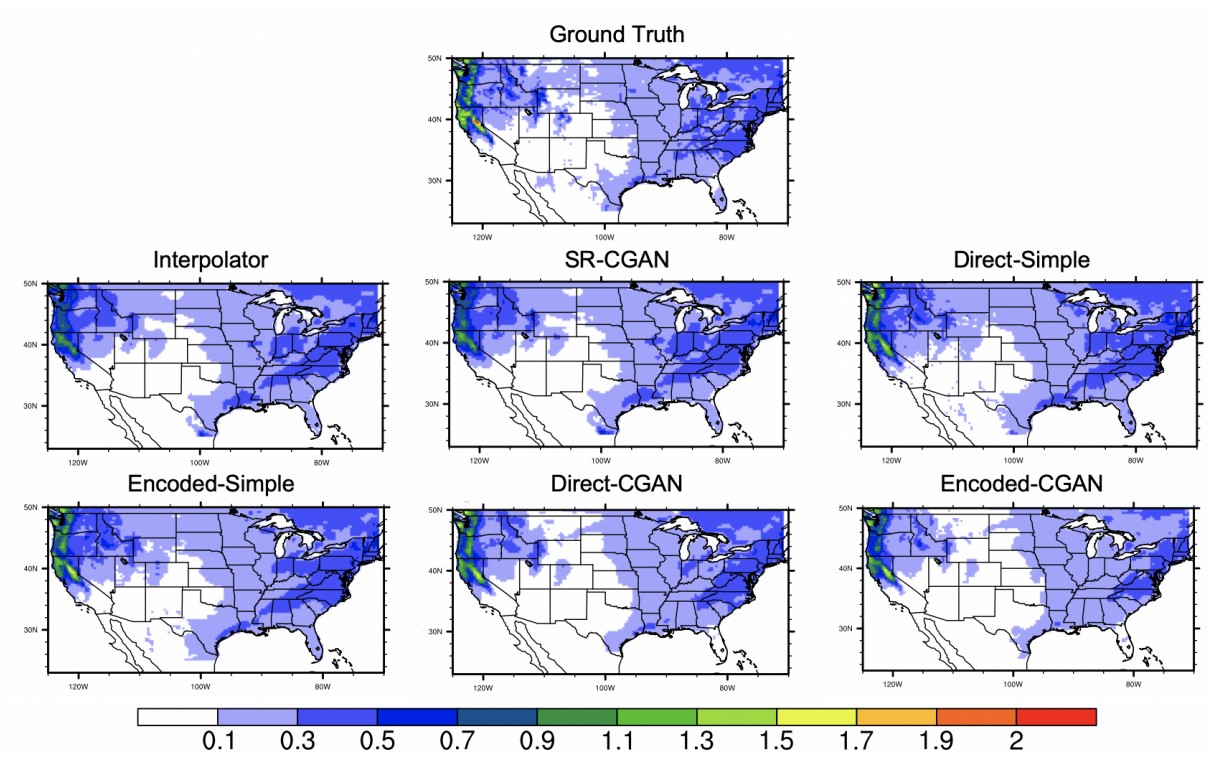

**Figure 7.** Mean precipitation (mm/3 hr) produced by Ground Truth, Interpolator, and five CNN models for the testing period (October–December).

dry. All CNN models capture the geospatial pattern of standard deviation, with the largest value over the northwestern coast, followed by moderate value over the eastern United States and the smallest value over Southwest. However, Interpolator and SR-CGAN show similar patterns (with pattern correlations with Ground Truth of 0.845 and 0.836, respectively) and do not have



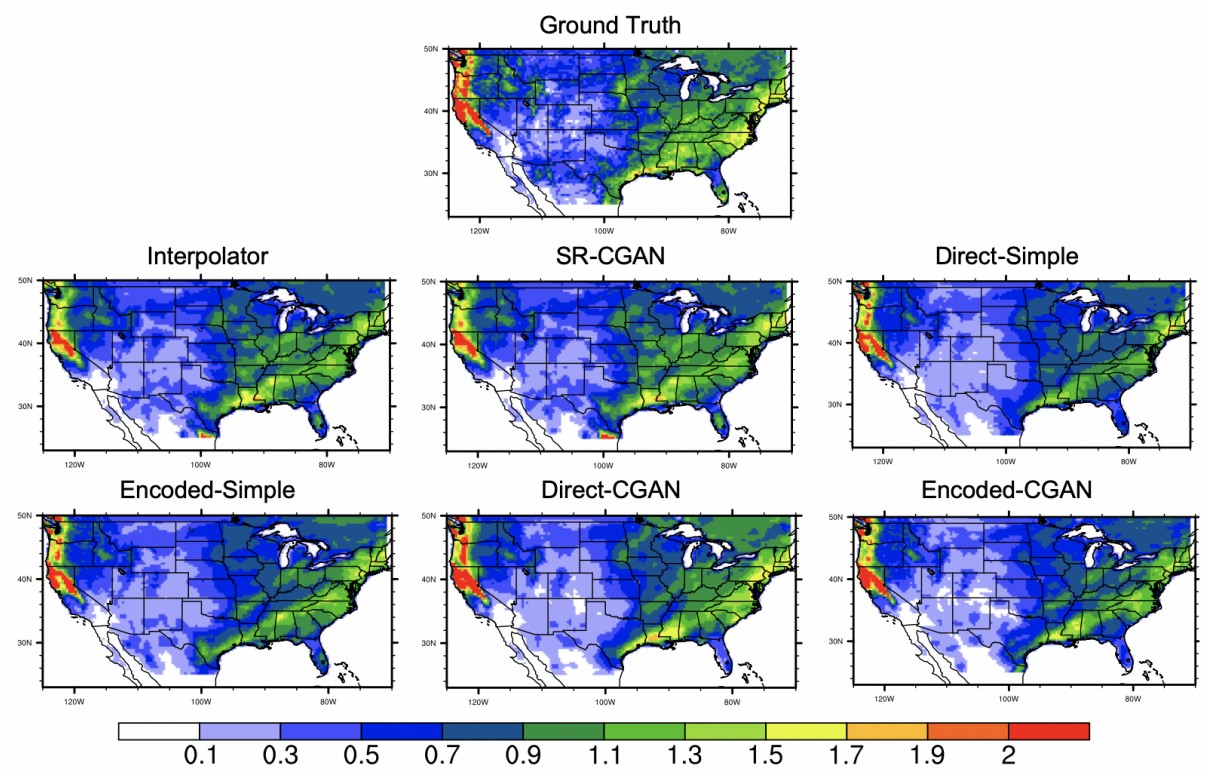

**Figure 8.** Standard deviation of precipitation (mm/3 hr) produced by Ground Truth, Interpolator, and five CNN models for the testing period (October–December).

values as large as shown in Ground Truth over the very northwestern coast, especially along the western coast of Washington and Oregon. The four new CNN models developed in this study improve the standard deviation over Northwest. The pattern correlation between Ground Truth and four CNN models are also improved to greater than 0.9 (Table 4), indicating that the new CNN models capture not only the precipitation variability along the time over each grid cell but also the spatial variability of the standard deviation.

Figure 9 shows the top 5% precipitation (averaged across 95th percentile to the maximum) during the testing period over each grid cell. The northwestern coast of Washington and Oregon and northern California have heavy precipitation, up to 10 mm/3 hr, and some southern states as well as the East Coast have precipitation up to 7 mm/3 hr. Interpolator and SR-CGAN underestimate the large precipitation over both the northwestern and eastern United States. All four CNN models developed here improve the precipitation amount over northern California and over western Oregon and Washington. The spatial variability of the top 5% precipitation is also improved by the four new CNN models, with pattern correlation increases from 0.861 by Interpolator to 0.92–93 by the new CNN models.

We also study the geospatial pattern of the 70th to 99th percentiles of all the CNN-predicted precipitation by comparing with Ground Truth and Interpolator. The new CNNs consistently outperform Interpolator and SR-CGAN and perform well for

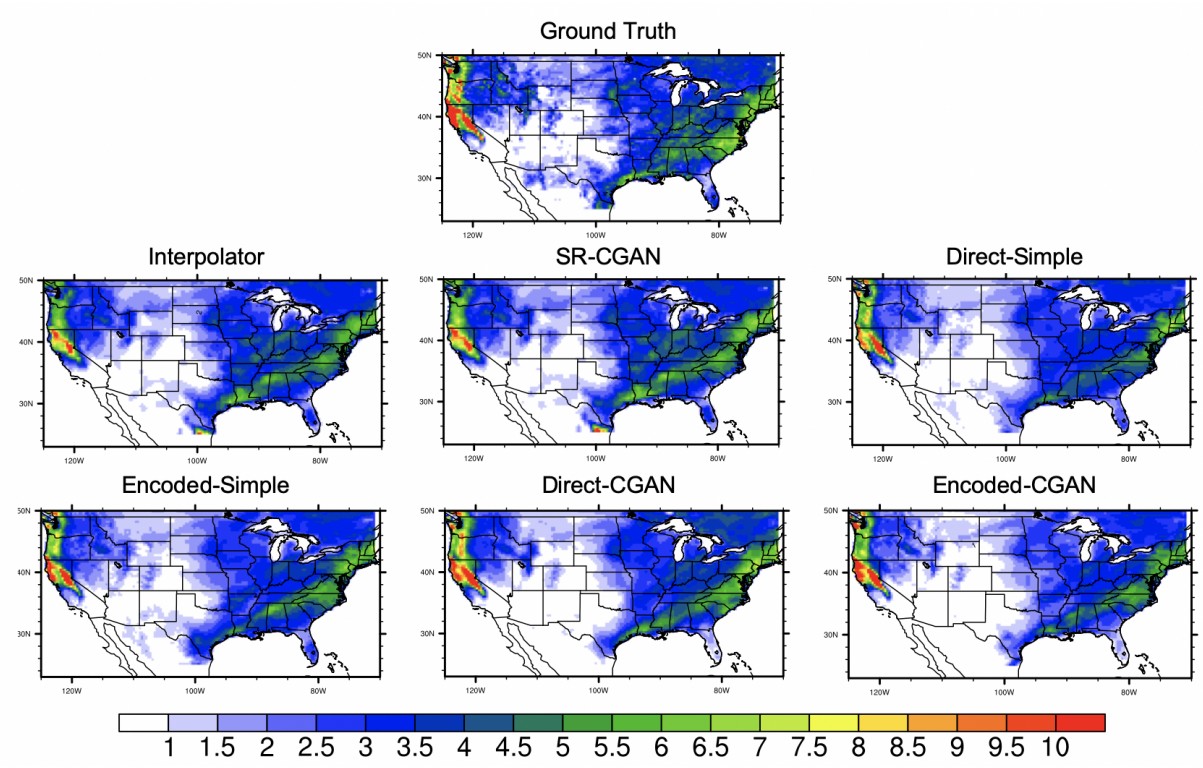

**Figure 9.** Top 5% (averaged across the 95th percentile to maximum) of the precipitation amount (mm/3 hr) during the testing period (October–December).

increasingly extreme precipitation events, except for the most extreme (>97th). Compared with Interpolator, the Simple models clearly reduce the MSE and improve the pattern correlations, especially for percentiles higher than the 80th. In particular, Encoded-Simple outperforms Direct-Simple. Direct-CGAN and Encoded-CGAN further improve the pattern correlations over the Simple models and show the best match to the Ground Truth in terms of the spatial variabilites of the extreme precipitation.

### 3.4 Event-Based Precipitation Characteristics

We investigate how well our different CNN models do at identifying and tracking precipitation events. To focus on events that are relevant to actual impact, we identify and track the precipitation events after removing grid cells with precipitation less than 10 mm/3 hr. First, we count the number of such events produced by each model during the testing period. We find 148 events in Ground Truth, 43 events in Interpolator, 57 events in SR-CGAN, 33 events in Direct-Simple, 69 events in Encoded-Simple, 57 events in Direct-CGAN, and 84 events in Encoded-CGAN. Thus, while Encoded-CGAN does the best, all models greatly underestimate the number of events.

To examine how well different models capture the characteristics of the storms seen in Ground Truth, we track the life cycle of storm events in each CNN-prediction to determine the total volume, duration, lifetime mean size, and lifetime mean intensity





**Table 4.** Pattern correlations between Ground Truth and six predictive models (Interpolator and the CNN-based models).

|  | Mean | Std. Dev | Top 5 |
|---|---|---|---|
| Interpolator | 0.866 | 0.845 | 0.861 |
| SR-CGAN | 0.871 | 0.836 | 0.854 |
| Direct-Simple | 0.937 | 0.915 | 0.928 |
| Encoded-Simple | 0.945 | 0.916 | 0.929 |
| Direct-CGAN | 0.943 | 0.903 | 0.922 |
| Encoded-CGAN | 0.951 | 0.912 | 0.930 |

of each event. We bin each of these characteristics and calculate their frequencies, giving the results in Figure 10. Looking first at precipitation intensity, we see that Ground Truth has intense precipitation events at greater than 11 mm/3 hr. While not captured by Interpolator, SR-CGAN, or Direct-Simple, this phenomenon is captured by Encoded-Simple, and Encoded-CGAN. This finding again indicates that the new CNN-based downscaling approach, especially when trained by CGAN, is useful for generating intense precipitation, while Interpolator and the state-of-the-art SR-CGAN tend to generate weaker precipitation
events and cannot capture these strong events sufficiently.

Looking next at duration, we see that while all the models capture the most frequent short-term events (0–3 hours), SR-CGAN and Direct-Simple are not able to capture the longer-term events. For lifetime mean size, the new CNN models tend to produce more larger events but fewer smaller events than are seen in Ground Truth. Looking at total volume of precipitation, we see that because the four CNN models tend to have more intense, larger, and longer-duration events than Interpolator and
SR-CGAN have, they show larger precipitation volumes more frequently than do Interpolator and SR-CGAN, and overall they better capture the large-volume precipitation events in Ground Truth. However, the four new CNN models—and, in particular, the CGAN models—do not show as frequent smaller-volume precipitation events as seen in Ground Truth. These small-volume precipitation events are well captured in Interpolator and SR-CGAN.

## 4 Summary and Discussion

This study develops a new CNN-based approach for downscaling precipitation from coarse spatial resolution RCMs. The downscaling approach is not constrained by the availability of observational data and can be applied to coarse-resolution simulation outputs to generate high-resolution precipitation maps with statistical properties (e.g., quantiles, including extremes and data variability) comparable to those seen in high-resolution RCM outputs. Because both the coarse-resolution simulations and neural network inferences are relatively inexpensive, our approach can greatly accelerate the process of generating such
simulated high-resolution precipitation maps.

Our approach is different from the super-resolution approach to downscaling taken by previous studies. Our CNNs are developed by using two datasets that are generated by two sets of RCM simulations run at different spatial resolutions. The resulting





**Figure 10.** Relative frequency (as %) of certain event-based precipitation characteristics: (a) lifetime mean intensity, mm/3 hr; (b) duration in (3-hr) timesteps; (c) lifetime mean size, km$^2$ (x-axis is in log); and (d) total volume, m$^3$ (x-axis is in log) during event lifetime.





precipitation images differ not only in resolution but also sometimes in geospatial patterns. One of the reasons is that the high resolution modeling handles the topographic and scale-dependent physical processes (e.g., resolved scale convection, boundary layer phenomena, and parameterized clouds) better than does the low resolution modeling. In addition, the slightly different model domain coverage and the significantly different computing timesteps can cause differences in precipitation fields. To mitigate the challenge of learning the relationship between the low- and high-resolution simulations when precipitation occurs in different locations, we use an inception module in the neural network to learn information from not only a target grid cell but also its surroundings (so-called receptive fields) in previous layers and generate data for that particular grid cell for the next layer. In addition, we employ the CGAN framework, which can help the CNN generate more physically realistic small-scale features and sharp gradients for precipitation in space.

We compare the new CNN-derived precpitation with precpitation generated from a Interpolator, which simply performs bilinear interpolation from coarse- to fine-resolution data, and that from the state-of-the-art SR-CGAN against Ground Truth. While Interpolator and SR-CGAN perform similarly, they are not as accurate as any of the four CNN methods when compared with Ground Truth, with overly smooth spatial precipitation patterns and underestimation of heavy precipitation. In contrast, the two new CGAN-trained CNNs produce the desired heavy-tailed shape, contributed by more intense and longer-lasting precipitation events that are in much better agreement with Ground Truth. In particular, when the CNN encoder is applied to the input variables, the output more accurately captures the spatial variabilities. These findings indicate that simply interpolating the coarse resolution or even using the state-of-the-art SR technique to generate fine-resolution data cannot capture the statistical distribution of precipitation.

The capability of generating high-resolution precipitation by using the technique developed in this study immediately suggests several interesting uses. For example, we can apply the CNN models to the outputs from the North American Regional Climate Change Assessment Program (NARCCAP; Mearns et al., 2012) or North American Coordinated Regional Climate Downscaling Experiments (NA-CORDEX; Mearns et al., 2017), both of which comprise multimodel 50 km WRF ensembles, to generate high-resolution precipitation data for uncertainty quantification in future projections. We also believe that the CNN models can be used to downscale output from a different RCM than that used to train the models, if that other RCM uses similar principal governing equations for simulated precipitation.

Although the CNNs that we have described here show promise, several limitations remain to be addressed. For example, we train the CNN with just nine months of data and test on the other three months of the same year. If we can train the CNN with multiple years of data that more fully capture the interannual variability, the algorithm might perform more robustly when applied to a new dataset. Another limitation is that our current CNN architecture does not consider dependencies between timesteps; instead, it processes images for each timestep independently. Therefore, the CNN output cannot capture temporal (here, 3-hourly) variations in precipitation data. However, the CNNs do capture the overall data variability and extremes over each grid cell, with improvements compared with Interpolator. Thus, peaks occurring at certain times in Ground Truth may not be captured by the CNN predictions, but the CNNs may have peaks with similar magnitudes at other times. While this performance is not satisfied in weather forecasting, it is acceptable for climate-scale simulations. In fact, in climate science the preference is to compare the statistical distribution of weather events (e.g., climate) rather than actual day-to-day weather.

Nevertheless, time dependencies in data can be important; weather propagates in time, and ideally precipitation images should not be treated independently.

Other studies have used a recurrent neural network structure (Leinonen et al., 2020) to permit generated outputs to evolve in time in a consistent manner, so that the GAN generator can model the time evolution of fields and the discriminator can evaluate the plausibility of image sequences rather than single images. The 3-hourly data that we use in this study are potentially too coarse to consider time dependencies: short-duration events may disappear between timesteps, and even for long-duration events, 3 hours may be too long to capture a smooth transition from one timestep to the next, as preferred by the learning

process. The other challenge is that once the time dimension is considered, the matrix will be three dimensional, which requires significantly larger computer memory. We plan to explore higher time-frequency dynamical downscaling simulations on more advanced GPU machines.

*Code and data availability.*   Source code is available at https://github.com/lzhengchun/dsgan. The data used in this study are available at http://doi.org/10.5281/zenodo.4298978

*Author contributions.*   JW participated in the entire project by providing domain expertise and analyzing the results from the CNNs. ZL designed, developed, and conducted all CNN experiments. WC conducted event-based analysis. IF, RK, and VRK proposed the idea of this project and provided high-level guidance and insight for the entire study.

*Competing interests.*   The authors declare that they have no competing interests.

*Acknowledgements.*   This material is based upon work supported by the U.S. Department of Energy, Office of Science, under contract
DE-AC02-06CH11357, and was supported by a Laboratory Directed Research and Development (LDRD) Program at Argonne National Laboratory through U.S. Department of Energy (DOE) contract DE-AC02-06CH11357.





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
