# Peer review of "Fast and accurate learned multiresolution dynamical downscaling for precipitation"

_Geoscientific Model Development, 2020_

## Referee Comment (RC1)

**Summary of the main contribution**

This manuscript proposes a novel deep learning approach to perform precipitation downscaling. Specifically, the authors do this by training five different CNN models (SR-CGAN, Directed/Encoded-Simple, Direct/Encoded-CGAN) to both low- and high- resolution (i.e., 50km and 12km horizontal resolution, respectively) Weather Research and Forecasting (WRF) simulations. The authors apply their methods to a one year WRF simulation and they assess the performance in terms of MSE, probability density function, spatial pattern of some selected summary statistics, and event-based rainfall intensity, duration, size, and total volume. Overall, I found the paper to be well motivated and most of it to be well described. However, I do have some concerns about the evaluation metrics used in this work.

**Major points**

My major concern has to do with some of the evaluation metrics used in this work:

1. **MSE**: I am not quite sure why the authors define MSE as $\frac{1}{N}\sum_{i=1}^{N}(Y_i - \bar{Y})^2$, where $N$ is the total number of grids, $Y_i$ is the (fitted) prediction value at grid $i$, and $\bar{Y}$ is the average prediction across all the grids. A more sensible MSE would be $\frac{1}{N}\sum_{i=1}^{N}(Y_i - Y_{i,\text{Groud Truth}})^2$. Also, I am not sure there is a need to calculate MSE at each timestep unless the authors plan to explore how MSE varies with time. Therefore I would suggest the authors to calculate MSE as $\frac{1}{N\times T}\sum_{i=1}^{N}\sum_{t=1}^{T}(Y_{i,t} - Y_{i,t,\text{Ground Truth}})^2$, where $T$ is the total time steps during the testing period.

2. **Precipitation Distribution:** I don't think J-S distance used here is very informative, a single number does not tell us *how* two distributions are different (and such assessment can be made, at least qualitatively, using log-pdfs shown in Fig. 6). However, it would be of interest to show spatial maps of J-S distance at each grid cell across different methods as it could potentially provide additional information for sub-region assessment in terms of fitted distributions. I would also suggest to replace the log-pdf curves in Fig. 6 by QQplots, that is, for each region, plot true (empirical) quantiles against the predicted quantiles. Quantile values are more directly interpretable than these log-density curves.

**Minor points**

⋆ page 2, line 42-43 *"Running an RCM is computationally expensive, however, and typically cannot be applied to large ESM ensembles"*:

   Please include reference for the Canadian Regional Climate Model Large Ensemble here, for example, Kirchmeier-Young, M. C., N. P. Gillett, F. W. Zwiers, A. J. Cannon, F. S. Anslow, 2018: Influence of human-induced climate change on British Columbias extreme 2017 fire season. Earth's Future, 7, 2-10. https://doi.org/10.1029/2018EF001050.

⋆ page 4, line 100 *"The data used in this study are one-year outputs..."*:

   Did the authors apply their methods to another one-year outputs to check if (qualitatively) similar conclusions can be obtained?

⋆ page 13, Table 2:

These information can be well summarized by a scatterplot by putting regions on x-axis and MSEs on y-axis with different color/line symbol combinations for these CNN models.

⋆ Figs. 7-9:

I would suggest the authors to plot relative error maps to better compare between different CNN model fits.

⋆ Fig 10:

I would recommend to use QQplots here for easier comparison.

---

## Referee Comment (RC2)

**Review of:**
Fast and accurate learned multiresolution dynamical downscaling for precipitation

**Recommendation:**
Major revisions

**Overview:**

The authors present a technique for dynamical downscaling using a neural network. They demonstrate that the CNN-based approach can significantly outperform interpolation-based downscaling and CNN-based super resolution (trained on statistically up-scaled data). The CNN-based downscaling approach is many times faster than traditional dynamical downscaling.

Overall, I think this is a significant contribution and am excited to see these CNN-based algorithms come into more widespread use. I do think the paper needs several major revisions. One is related to the MSE evaluation metric. The others generally ask for more exposition about methods and should be fairly easy to address. (Also, regarding methods, thank you to the authors for including a link to their code, this was helpful and contributes to the reproducibility of the work).

**Major Comments:**

S 2.1: How is scaling/pre-processing/post-processing of the data handled? It looks, based on the code, like the output activation from the CNN is a sigmoid (logistic) function, so how are the precipitation data mapped to/from the (0,1) range? Do you take any steps to account for the very skewed distribution of precipitation? Does the sigmoid output cap the maximum possible output precipitation? If there is a non-linear transform used to map from the CNN outputs back to dimensional precipitation values how does this affect using an MSE loss function? These considerations should be addressed in the manuscript.

Lines 138, 166 and Fig 3: It is not clear how the elevation data are passed to the CNN. Line 138 says it is concatenated to the low-res inputs and Line 166+Fig 3 make it seem that it is only concatenated to features derived from the other fields near the end of the network.

Eqns 1 and 2: Please explain what 'D' is other than just the discriminator output. Your code looks like it is binary cross entropy applied to a sigmoid output with binary class labels. This info should be included here.

Eq 3. Does not seem like an appropriate error metric for this problem. Why is the difference between the SR/downscaling and a spatial mean of precipitation used? Shouldn't this be something like: MSE = $(1/N) \sum_{i=1}^N (Y_i^{CNN} - Y_i^{GT})$? (GT = ground truth). I believe that is the type of MSE the CNN is optimizing. To me, Eq 3 looks more like the variance of the CNN output computed about the ground truth mean rather than the MSE. Also, just

practically speaking, I don't think adequately estimating total precipitation averaged over all of conus really matters if we can't accurately say where within CONUS it is happening.

S 2.5: No information about the discriminator network architecture is given.

S 3.2: What are the relative numbers of trainable parameters between the models? Do the encoded CNNs improve performance (in terms of the PDFs at least) because of the model architecture difference or simply because they have more convolutional kernels?

**Minor Comments:**

Title: what is meant by "multi-resolution"? doesn't this only produce 12km outputs?

General: There's inconsistent terminology used in the ML and atmospheric science fields: "up-sampling" and "down-scaling" both refer to resolution increases while "down-sampling" and "up-scaling" both refer to resolution decreases. It would be helpful to make this clear somewhere in the manuscript to prevent confusion for a reader who is only familiar with one of the fields.

Abstract/Line 73: When reading the abstract, it's certainly implied but I don't think it is explicitly stated that a key difference between your method and the CNN-SR method is that one uses high- and low-resolution simulations to train and the other uses a high-res simulation with statistical up-scaling to train.

Line 93: "precipitation images" – is an odd term since the data are really only similar to an image in that they are gridded/raster data. Presumably the CNN is operating on the actual precipitation data and not the RGB images of it.

Line 140: again I don't think referring to the data as images is helpful since they are not images. Something like: "This approach of stacking the variables as different input channels has been used in other downscaling studies…" would be fine.

Line 138: *A friendly suggestion regarding the elevation data*
If you are only concatenating the DEM data near the end of the network like Fig 3 seems to suggest, you may want to consider adding the elevation data earlier in the model. It appears that after the DEM data are concatenated they are only passed through one layer alongside the features learned from the other inputs before the model output. I suspect this could limit the CNN's ability to learn the complicated non-linear relationships between the DEM and the other fields needed to estimate orographic precipitation (letting the information about orography flow through the channel attention blocks might be particularly helpful). I would recommend finding a way to add the DEM data earlier in the network, perhaps you could use convolutional downsampling or a pixel shuffle instead of just 2d-avg to get it down to the same resolution as

the other inputs without losing much information. I think the paper is fine without doing this of course, since you have already achieved very good performance.

S 2.3: Did you provide any source of random variability to the CGAN?

Table 2: Units?

SS2.5 It feels disorganized to introduce the GAN model in a section labeled "loss functions." I suggest just having a section early on that briefly introduces each model to help the reader keep everything straight. Then proceed to discuss specifics of implementation / loss functions etc.

**Very Minor / Typos:**

Line 10: Awkward sentence

Line 21: change to something like: "ESMs cannot *fully* resolve cloud processes." the way it's currently worded makes it sound like the models don't include clouds at all

Line 45: one number is formatted with a comma and one without

Line 57: GAN-based SR doesn't necessarily improve pixel accuracy over a conventional CNN, they improve feature loss or realism. Maybe just be clear about what you mean by "accuracy"

109: "convections"→"convection"

F1: Thanks for including this diagram, very helpful

L 194-195: I don't think this sentence is correct

L 204: "less good" → "worse"

L 225: wow that's fast! I'm excited to see algorithms like this used operationally

---

## Author Response (AR1)

Dear Editor and Reviewers,

We greatly appreciate your handling of our manuscript, *Fast and accurate learned multiresolution dynamical downscaling for precipitation.* We especially thank the two anonymous referees for their detailed and constructive comments and suggestions, which helped improving the manuscript significantly. To respond to reviewers' comments and suggestions, we mark all the changes in purple in our revised manuscript. Major efforts include that, we made quantile-quantile (q-q) plots; we made J-S distance maps, we updated Figs 7-9 with relative error maps; we conducted another experiment to increase the number of parameters in Direct models for investigating whether the Encoded models' improvement is due to model size or model architecture; we have also cleaned up our source code and added more comments. In addition, we updated Figure 3 and the numbers in Table 2. Please find our one-on-one responses below to the comments and suggestions provided by the reviewers.

Sincerely,

Jiali Wang, Zhengchun Liu, Ian Foster, Won Chang, Rajkumar Kettimuthu, Rao Kotamarthi.

**1 Comments and Response to Reviewer #1**

**Comments**: MSE: I am not quite sure why the authors define MSE as $\frac{1}{N}\sum_{i=1}^{N}\left(Y_i - \overline{Y}\right)^2$, where N is the total grids. A more sensible MSE would be $\frac{1}{N}\sum_{i=1}^{N}(Y_i - Y_{i,GroundTruth})$. Also, I am not sure there is a need to calculate MSE at each timestep unless the authors plan to explore how MSE varies with time. Therefore I would suggest the authors to calculate MSE as $\frac{1}{N \times T}\sum_{i=1}^{N}\sum_{t=1}^{T}(Y_{i,t} - Y_{i,t,GroundTruth})^2$ where T is the total time steps during the testing period.

**Response**: Thanks for pointing this out. We have revised the equation for MSE (eq. 2) in Page 11, as suggested. We computed MSE for each time-step, because we assess the performance of our deep-learning (DL) models in different percentiles of the precipitation. For example, in Table 2, we present the MSE for the 50th and 99th percentiles of precipitation. We agree that, if the variation of MSE with time is not of an interest, then the MSE could be calculated for all grid cells at all times. Here is what we have in the revised manuscript:

$$\ell_{mse} = \frac{1}{N}\sum_{i=1}^{N}\left(Y_i^H - G_\theta\left(Y_i^L\right)\right)^2, \tag{1}$$

where $N$ is the total number of grid cells over the study domain, $Y_i^H$ and $Y_i^L$ are the precipitation at grid cell ($i$) simulated by WRF at high (12 km) and low (50 km) resolution respectively. $Y_i^H$ is used as Ground Truth in this case. $G_\theta$ is the deep neural network parameterized with $\theta$ that models the difference between low and high resolution simulations.
* * *
**Comments**: Precipitation Distribution: I don't think J-S distance used here is very informative, a single number does not tell us how two distributions are different (and such assessment can be made, at least qualitatively, using log-pdfs shown in Fig. 6). However, it would be of interest to

[Figure]

Figure R1: J-S distance measuring the similarity of the PDFs between Ground Truth and six predictive models for the testing period (October–December).

show spatial maps of J-S distance at each grid cell across different methods as it could potentially provide additional information for sub-region assessment in terms of fitted distributions. I would also suggest to replace the log-pdf curves in Fig. 6 by QQplots, that is, for each region, plot true (empirical) quantiles against the predicted quantiles. Quantile values are more directly interpretable than these log-density curves.

**Response**: According to reviewer's suggestions, (1) we have added the J-S distance maps which show the similarity of the PDFs of each grid cell between Ground Truth and the six different downscaling approaches. As shown in Figure R1, compare with Interpolator and SR-CGAN, while the Simple models produce the largest distances from the Ground Truth, the Encoded-CGAN produces the smallest distances over the entire CONUS, indicating that Encoded-CGAN can generate closer precipiation distributions to Ground Truth than the other five downscaling approaches.

(2) We have made Q-Q plots using all the non-zero precipitation data and the precipitation data greater than 8mm/3hr over each of the seven subregions. We found that, the Q-Q plot using all non-zero precipitation is dominated by the majority of the data which are not really at the tails; while the Q-Q plot using precipitation greater than 8mm/3hr can capture the information at the tail, and we also found them showing a similar conclusion as the PDF plots (see Figure R2). That is, the CGAN models usually can capture the very extremes better than the other downscaling approaches. Since we are interested in evaluating the tails of the distributions predicted by different DL models, the PDF plots are more suitable than Q-Q plot. For example, if there are two distributions with similar tails, but they are different in other parts of the distribution, then Q-Q plot may conclude that these two distributions are very different (see Figure R3). Therefore, we kept our PDF plots and the J-S distances (Table 3) that are used to quantify the similarity in PDFs generated by different DL models.

[Figure]

Figure R2: PDF (top) and Q-Q (bottom) plots over Midwest and Northwest. In PDF plots, the red line with cross is Encoded-CGAN, and the blue line with triangles is Encoded-Simple. Both PDF and Q-Q plots indicate that CGAN models are closer to Ground Truth over Midwest, and Encoded models are closer to Ground Truth over Northwest.

**Comments**: * page 2, line 42-43 "Running an RCM is computationally expensive, however, and typically cannot be applied to large ESM ensembles": Please include reference for the Canadian Re-

[Figure]

Figure R3: An example of two distributions plotted in histogram and Q-Q plots. These two distributions are similar on the right side (greater than 0) but is different on the left side.

gional Climate Model Large Ensemble here, for example, Kirchmeier-Young, M. C., N. P. Gillett, F. W. Zwiers, A. J. Cannon, F. S. Anslow, 2018: Influence of human-induced climate change on British Columbias extreme 2017 fire season. Earth's Future, 7, 2-10. https://doi.org/10.1029/2018EF001050.

**Response**: Thanks and the citation is included.
* * *
**Comments**: * page 4, line 100 "The data used in this study are one-year outputs...": Did the authors apply their methods to another one-year outputs to check if (qualitatively) similar conclusions can be obtained?

**Response**: The neural networks we developed are only based on one year of data which is not able to capture the inter-annual variability. Thus when the trained model is applied to a different year, we expect the model performance could be unsatisfied. On the other hand, if we conduct both training and testing for a different year (e.g., use first 9 months for training data, and the rest for testing), we expect the conclusion will be similar to what we got here. That is, our multi-resolution model can produce high-resolution modeled precipitation data with comparable statistical properties but at greatly reduced computational cost. This training and verification for a new year will take significant amount of effort as we did before with the data from year 2005. It is possible though, that we may end up with a slightly different set of hyperparameters that achieves the best results. This will require careful tuning and testing of the model development. With more computational resource available in near future, we would like to train our neural networks with multiple years of data to consider the inter-annual variability; we also would like to apply our techniques to other coarse resolution models, such as the state-of-the-art earth system models that are on grid spacing of 50-100 km. We added this discussion in the section of Summary and Discussion.
* * *
**Comments**: * page 13, Table 2: These information can be well summarized by a scatter plot by

putting regions on x-axis and MSEs on y-axis with different color/line symbol combinations for these CNN models.

[Figure]

Figure R4: MSEs, calculated across all grid cells over the entire CONUS and seven subregions, at the 50th and 99th percentiles picked from all timesteps.

**Response**: Thanks for the suggestion. We have tried the scatter plots, as shown by Figure R4. We can see that, because the MSEs are larger over Northwest and CONUS than those over the other six subregions, putting them in one plot make it difficult to read the differences between different downscaling approaches. We thus kept the original table, but we updated the numbers in Table 2 as we found some numbers are incorrect. In addition, we would like to emphasize that, in Table 2 we can see that MSE does not seem to be a sufficient metric for evaluating precipitation, especially its spatial patterns, because MSE tends to evaluate the spatial average instead of the small-scale features (e.g., heavy precipitation), which is more important in real-situation applications such as risk assessment of heavy precipitation or flooding.
* * *
**Comments**: Figs. 7-9: I would suggest the authors to plot relative error maps to better compare between different CNN model fits.

**Response**: Thanks for the suggestion. We have updated these figures (Figs. 8-10 in revised manuscript) with relative error maps. An example of the difference in top 5% precipitation between Ground Truth and all six downscaling approaches is shown by Figure R5. We can see that both Interpolator and SR-CGAN underestimate the high precipitation over Northwest, but overestimate the precipitation over central and eastern US, indicating smoother spatial patterns than those generated by the four DL models we developed. Both the Simple models and the CGAN models reduce the bias (e.g., underestimation shown in Interpolator and SR-CGAN) over Northwest and over south central and Southeast (e.g., overestimation shown in Interpolator and SR-CGAN). We see similar improvements for standard deviation and monthly averaged precipitation.
* * *
**Comments**: * Fig 10: I would recommend to use QQplots here for easier comparison.

[Figure]

Figure R5: Differences in top 5% (averaged across the 95th percentile to maximum) of the precipitation amount (mm/3 hr) during the testing period (October–December).

**Response**: As we responded to the previous comment about Q-Q plots, because the distributions of the precipitation features (intensity, size, duration, and frequency) from each DL model are not exactly the same, and we are interested more in the extremes, we kept the histogram plots for each of the characteristic for the rainstorm. Please note here we only tracked the events that have precipitation amount greater than 10mm/3hr, because we focus on the very extremes due to their large impacts. It is the same reasons that we use PDF plots for model evaluations for the extremes.

**2 Comments and Response to Reviewer #2**

**Comments**: How is scaling/pre-processing/post-processing of the data handled? It looks, based on the code, like the output activation from the CNN is a sigmoid (logistic) function, so how are the precipitation data mapped to/from the (0,1) range? Do you take any steps to account for the very skewed distribution of precipitation? Does the sigmoid output cap the maximum possible output precipitation? If there is a non-linear transform used to map from the CNN outputs back to dimensional precipitation values how does this affect using an MSE loss function? These considerations should be addressed in the manuscript.

**Response**: Thanks for your questions. The sigmoid function you saw in the source code is for the spatial and channel attention module, i.e., the CBAM as illustrated in Fig. 3. The activation for the output layer of the generator is linear. The sigmoid function is not for precipitation data, and we did not map precipitation to the (0,1) range. To avoid confusion, we added comments in the source code to clarify the coding structure at the first place. We have also renamed several functions to make them more readable. Please find the refined source code available in the same repository `https://github.com/lzhengchun/DSGAN`.
* * *
**Comments**: Lines 138, 166 and Fig 3: It is not clear how the elevation data are passed to the CNN. Line 138 says it is concatenated to the low-res inputs and Line 166+Fig 3 make it seem that it is only concatenated to features derived from the other fields near the end of the network.

**Comments**: Line 138: *A friendly suggestion regarding the elevation data* If you are only concatenating the DEM data near the end of the network like Fig 3 seems to suggest, you may want to consider adding the elevation data earlier in the model. It appears that after the DEM data are concatenated they are only passed through one layer alongside the features learned from the other inputs before the model output. I suspect this could limit the CNN's ability to learn the complicated non-linear relationships between the DEM and the other fields needed to estimate orographic precipitation (letting the information about orography flow through the channel attention blocks might be particularly helpful). I would recommend finding a way to add the DEM data earlier in the network, perhaps you could use convolutional downsampling or a pixel shuffle instead of just 2d-avg to get it down to the same resolution as the other inputs without losing much information. I think the paper is fine without doing this of course, since you have already achieved very good performance.

**Response**: Thanks for your comment and suggestion. Here we are responding them together since they are referring the same figure (Figure 3) about the elevation or DEM data. We realize that our Figure 3 did not fully represent our actual implementation – thanks for pointing it out! As you may have seen in our open source code, the elevation data is concatenated to features after two upsampling operations when the width and height of features match the elevation data. After the concatenating the elevation and the other variables (T2, IWV and SLP) at 12 km, we actually have four inception boxes before the output layer. We have updated Figure 3 in the revised manuscript. It is also shown by Figure R6.

We have also corrected Line 138 by deleting the 'elevation'. It reads now "we directly stack all

[Figure]

Figure R6: Model architecture for Direct-Simple and the generator of Direct-CGAN. CBAM=Convolutional Block Attention Module. TConv=Transposed convolution.

selected variables (precipitation, T2, IWV, SLP) to form a three-dimensional tensor as input to the CNN model;"
* * *
**Comments**: Eqs 1 and 2: Please explain what 'D' is other than just the discriminator output. Your code looks like it is binary cross entropy applied to a sigmoid output with binary class labels. This info should be included here.

**Comments**: S 2.5: No information about the discriminator network architecture is given.

**Response**: We are responding these two comments together since they are both about the discriminator. Let `CxKySz` denote a convolution layer with `x` channels, kernel size of `y` and striding of `z`. Our discriminator network architecture is `C64K4S2-C128K4S2-C256K4S2-C512K4S2-C512K4S2-C1K4S2`. Batch Normalization are applied for outputs of layer 2 to layer 5. We used leaky ReLU activation function with a negative slope of 0.2 for all layers except the last layer (the output layer) for which the sigmoid activation is used. In the revised manuscript, we added the suggested sentence here: "$D$ is the discriminator, and is binary cross entropy applied to a sigmoid output with binary class labels."
* * *
**Comments**: Eq 3. Does not seem like an appropriate error metric for this problem. Why is the difference between the SR/downscaling and a spatial mean of precipitation used? Shouldn't this be something like: MSE $= (1/N) \sum_{i=1}^{N} (Y_i^{CNN} - Y_i^{GT})$? (GT = ground truth). I believe that is the type of MSE the CNN is optimizing. To me, Eq 3 looks more like the variance of the CNN output computed about the ground truth mean rather than the MSE. Also, just practically speaking, I don't think adequately estimating total precipitation averaged over all of conus really matters if we can't accurately say where within CONUS it is happening.

**Response**: Thanks for pointing this out. We have corrected eq. 3 as suggested by both reviewers. We also agree that, it is very important that a model can predict the precipitation over specific regions or locations. To evaluate this regard, we conducted model evaluation over each of the subregions, and also over each grid cell by looking at geospatial patterns across the entire CONUS in Section 3.3 Geospatial analysis of other measures, illustrated by Figures 7-10.
* * *
**Comments**: S 3.2: What are the relative numbers of trainable parameters between the models?

Do the encoded CNNs improve performance (in terms of the PDFs at least) because of the model architecture difference or simply because they have more convolutional kernels?

**Response**: The number of parameters for the encoded-CGAN (1,487,683 trainable parameters) is a little (2.74%) more than the direct-CGAN (1,447,971 trainable parameters) as encoded-CNN has the encoder layers shown in Fig. 5. In order to investigate whether the improvement is due to the larger number of parameters or the neural network architecture, we conducted an experiment by adding more inception boxes (the encoder also uses the inception box) to the direct-CGAN to make the parameters similar as the encoded-CGAN. It ended up with 1,499,363 parameters, a little more than the encoded-CGAN. Let's name the new model as Direct-Big-CGAN. The median MSE of all grid cells (as shown in 1st column of Table 2) are almost identical between the Direct-Big-CGAN and the original direct-CGAN (0.21043 vs. 0.21015).This result indicates that even with a larger number of parameters, the performance of the direct-CGAN is still the same. Therefore, the improvements of encoded-CGAN compared with direct-CGAN over many subregions for PDFs (shown in J-S distance maps) are most likely due to the neural network architecture not the larger number of parameters. We added this discussion to the section of Summary and Discussion.
* * *
**Comments**: Title: what is meant by "multi-resolution"? doesn't this only produce 12km outputs?

**Response**: Our output from the deep learning based downscaling approach only produce 12km outputs for this particular study, you are correct. However, by "multi-resolution", we mean that our approach is developed based on datasets that are on more than one spatial resolution. This is contrast to the state-of-the-art Super-Resolution technique, which upsample a high resolution data, and develop the SR model using the upsampled low and high resolution data which are based on the same data source. The strength of our approach is that during the training, the neural network can model the difference between two completely different datasets that are generated by low and high resolution numerical simulations. This is particularly useful for climate model or earth system model output, because they can generate different output with different spatial resolutions. As we explained in the 2nd paragraph of Section 2.1, the difference between two simulations that are run at different spatial resolutions involve terrain effect, effects of time steps, impacts of slightly different model domain coverage. Therefore, modeling the difference between these two datasets is more challenging but can potentially produce better results when apply it to a low resolution model simulation. Note that this low resolution data was used during the training.
* * *
**Comments**: General: There's inconsistent terminology used in the ML and atmospheric science fields: "upsampling" and "down-scaling" both refer to resolution increases while "down-sampling" and "up-scaling" both refer to resolution decreases. It would be helpful to make this clear somewhere in the manuscript to prevent confusion for a reader who is only familiar with one of the fields.

**Response**: Thanks for the comment and agreed. We added a clarification to the end of Introduction. It reads "It is noteworthy that "upsampling" and "downscaling" both refer to resolution increases (from low to high resolution), while "downsampling" and "upscaling" both refer to resolution decreases (from high to low resolution). Since this is an interdisciplinary study, different

terms are used in different contexts but refer to the same meanings"
* * *
**Comments**: Abstract/Line 73: When reading the abstract, it's certainly implied but I don't think it is explicitly stated that a key difference between your method and the CNN-SR method is that one uses high- and low-resolution simulations to train and the other uses a high-res simulation with statistical up-scaling to train.

**Response**: Yes, that is the key difference, and that is also what we mean by "multi-resolution" in the title. We train DNNs to capture the difference between low- and high-resolution simulations. Their difference is not only in spatial resolution but also in geospatial patterns, as shown in Figure 2; CNN-SR, on the other hand, upscales a high-resolution image to low-resolution and then generate a high-resolution data that is close enough to the original high-resolution data. So the only difference in CNN-SR's training data is the spatial resolution. In the revised manuscript, we emphasize this regard in both Abstract and Line 73. Abstract reads now: "The key idea is to use combination of low- and high- resolution simulations (differ not only in spatial resolution but also in geospatial patterns) to train a neural network to map from the former to the latter.". Line 73 reads now: "Two datasets (i.e., low- and high-resolution simulations) rather than one (i.e., only high-resolution with its upscaling) are used to develop the new CNN-based downscaling approach."
* * *
**Comments**: Line 93: "precipitation images" – is an odd term since the data are really only similar to an image in that they are gridded/raster data. Presumably the CNN is operating on the actual precipitation data and not the RGB images of it.

**Response**: We agree with the reviewer and have changed the "precipitation images" to "precipitation visualization" as we basically compare it visually there. We also changed "precipitation images" in other contexts to "precipitation" as we do not really talk about image or visulization but just the precipitation data itself.
* * *
**Comments**: Line 140: again I don't think referring to the data as images is helpful since they are not images. Something like: "This approach of stacking the variables as different input channels has been used in other downscaling studies..." would be fine.

**Response**: Corrected.
* * *
**Comments**: S 2.3: Did you provide any source of random variability to the CGAN?

**Response**: By "random variability", do you mean the latent variable input to the generator? We do not provide this because as we mentioned in the manuscript, "..we use actual precipitation amounts and the conditional variables as inputs, forming a conditional GAN (CGAN) framework for training the generator.". The discriminator is a helper here to provide a better loss function for the generator.

**Comments**: Table 2: Units?

**Response**: These are MSE for precipitation amount at each time step. We use 3 hourly precipitation data, so the MSE here is in mm/3h. Added in Table 2 caption.
* * *
**Comments**: SS2.5 It feels disorganized to introduce the GAN model in a section labeled "loss functions." I suggest just having a section early on that briefly introduces each model to help the reader keep everything straight. Then proceed to discuss specifics of implementation / loss functions etc.

**Response**: Thanks for the suggestion. We first describe the model architecture, and they are Direct-Simple and Encoded-Simple models. Then we describe the loss functions we use, and they are MSE and CGAN. Accordingly, we introduce the models Direct-CGAN and Encoded-CGAN. So the first part of our model names is about architecture, and the second part is about loss function.Overall, we feel it flows well, so we kept the original structure.
* * *
**Comments**: Line 10: Awkward sentence

**Response**: Corrected.
* * *
**Comments**: Line 21: change to something like: "ESMs cannot fully resolve cloud processes." the way it's currently worded makes it sound like the models don't include clouds at all

**Response**: Revised, and it reads now: "Such resolutions are not sufficient to fully resolve critical physical processes such as clouds.."
* * *
**Comments**: Line 45: one number is formatted with a comma and one without

**Response**: Corrected.
* * *
**Comments**: Line 57: GAN-based SR does not necessarily improve pixel accuracy over a conventional CNN, they improve feature loss or realism. Maybe just be clear about what you mean by "accuracy"

**Response**: We added the clarification as suggested.
* * *
**Comments**: 109: "convections" → "convection"

**Response**: Corrected.
* * *
**Comments**: F1: Thanks for including this diagram, very helpful

**Response**: We are glad that the reviewer finds this figure helpful. Once again we would like to emphasize that, the difference between our study and the state-of-the-art Super-Resolution technique is that, we use multi-resolution datasets to train the DL models. Results show that it can decently capture the statistical features when comparing with 'Ground Truth' (high-resolution WRF simulation here).
* * *
**Comments**: L 194-195: I don't think this sentence is correct

**Response**: Can the reviewer please indicate the problem in this sentence? We are happy to discuss more.
* * *
**Comments**: L 204: "less good" → "worse"

**Response**: Corrected.
* * *
**Comments**: L 225: wow that's fast! I'm excited to see algorithms like this used operationally

**Response**: Thanks and we also find this information encouraging. The speed-up these DL models have achieved provide us opportunities to pursue high spatial resolution simulations as well as large ensemble member simulations to better quantify the uncertainties in the climate models. We sincerely thank the reviewers again for all your constructive comments and suggestions!